# Potentials of Presepsin as a Novel Sepsis Biomarker in Critically Ill Adults: Correlation Analysis with the Current Diagnostic Markers

**DOI:** 10.3390/diagnostics15020217

**Published:** 2025-01-18

**Authors:** Mai S. Sater, Nourah Almansour, Zainab Hasan Abdulla Malalla, Salim Fredericks, Muhalab E. Ali, Hayder A. Giha

**Affiliations:** 1Department of Medical Biochemistry, College of Medicine and Medical Sciences (CMMS), Arabian Gulf University (AGU), Manama 293, Bahrain; zainabhm@agu.edu.bh (Z.H.A.M.); muhalabae@agu.edu.bh (M.E.A.); 2Immunology and Microbiology Department, Dasman Diabetes Institute, Dasman 15462, Kuwait; nora-86-kwt@hotmail.com; 3Department of Biochemistry, Royal College of Surgeons in Ireland–Medical University of Bahrain (RCSI-MUB), Busaiteen 228, Bahrain; sfredericks@rcsi-mub.com; 4Medical Biochemistry and Molecular Biology, Khartoum, Sudan; gehaha2002@yahoo.com

**Keywords:** sepsis, critically ill patients, diagnosis, biomarkers, presepsin, cytokines

## Abstract

**Background:** Sepsis is a major cause of patient death in intensive care units (ICUs). Rapid diagnosis of sepsis assists in optimizing treatments and improves outcomes. Several biomarkers are employed to aid in the diagnosis, prognostication, severity grading, and sub-type discrimination of severe septic infections (SSIs), including current diagnostic parameters, hemostatic measures, and specific organ dysfunction markers. **Methods:** This study involved 129 critically ill adults categorized into three groups: sepsis (Se = 48), pneumonia (Pn = 48), and Se/Pn (33). Concentrations of five plasma markers (IL-6, IL-8, TREM1, uPAR, and presepsin) were compared with 13 well-established measures of SSI in critically ill patients. These measures were heart rate (HR), white blood count (WBC), C-reactive protein (CRP), procalcitonin (PCT), lactate plasma concentrations, and measures of hemostasis status (platelets count (PLT), fibrinogen, prothrombin time (PT), activated partial thromboplastin time (APTT), international normalization ratio (INR) and D-dimer). Plasma bilirubin and creatinine served as indicators of liver and kidney dysfunction, respectively. **Results:** Promising roles for these biomarkers were found. The best results were for presepsin, which scored 10/13, followed by IL-6 and IL-8 (each scored 7/13), and the worst were for TREM-1 and uPAR (scored 3/13). Presepsin, IL-6, and IL-8 discriminated between the SSI sub-types, whilst only presepsin correlated with bilirubin and creatinine. uPAR was positive for kidney dysfunction, and TREM-1 was the only indicator of artificial ventilation (AV). **Conclusions:** Presepsin is an important potential biomarker in SSIs. However, further work is needed to define this marker’s diagnostic and prognostic cutoff values.

## 1. Introduction

Sepsis is a life-threatening organ malfunction caused by a dysregulated host response to infection (Singer et al., 2016 [1]). The definition and diagnostic criteria of sepsis vary in different clinical settings. The World Health Organization (WHO) defines sepsis as a serious condition that happens when the body’s immune system has an extreme response to an infection, with consequent damage to its own tissues and organs. Common signs and symptoms of sepsis include fever or low body temperature and shivering, confusion, difficulty in breathing, clammy and sweaty skin, extreme body pain or discomfort, high heart rate (HR), weak pulse or low blood pressure (BP), and low urine output. It can lead to septic shock, multiple organ failure, and death. Generally, sepsis is caused by bacterial infections (WHO, 2023 [2]). However, infections with viruses, parasites, or fungi may also be causes of sepsis (WHO, 2023 [2]). The triad of infections, immunity, and inflammation is believed to initiate sepsis and sustain its progression. Sepsis may be localized to specific organs, as with pneumonia, or it may become systemic, as with septicemia. This condition potentially affects all bodily systems. The term sepsis is applicable to all types of bacterial infections; we therefore prefer the term severe septic infection (SSI) in the context of this study.

Sepsis is common among hospitalized patients, accounting for around 30–50% of hospital mortality (Angus et al., 2001 [3]; Liu et al., 2014 [4]). These deaths may, in part, be attributed to the unavailability of distinct and reliable SSI biomarkers for early diagnosis, and later progression of infection, with consequent delays in the commencement of treatment (Rhee et al., 2019 [5]). Increases in circulating procalcitonin (PCT), serum amyloid A (SAA), soluble triggering receptor expressed on myeloid cells-1 (sTREM-1), mannan, anti-mannan antibodies, interleukin 6 (IL-6), interleukin 8 (IL-8), monocyte chemoattractant protein-1 (MCP-1), presepsin, and soluble urokinase-type plasminogen activator receptor (suPAR) have all been associated with sepsis and been used for early detection and prognosis assessment (Larsen and Petersen, 2017 [6]).

The gold standard for diagnosing sepsis is a microbial culture in the laboratory to identify the cause of infection. However, this remains impractical, as laboratory results typically take 24–48 h to become available, thus impeding treatment initiation, which is a crucial step in managing sepsis (Wang et al., 2010 [7]). Moreover, it has been reported that as few as 54% of sepsis patients are found to have positive blood culture tests (Mellhammar et al., 2012 [8]). This poor sensitivity of the blood culture test raises doubts regarding the applicability of the term gold standard for blood culture. Without a single practical and appropriate definitive diagnostic test, there is a desperate need for markers as early indicators for SSIs. The use of non-specific markers such as serum C-reactive protein (CRP) and procalcitonin (PCT), along with the clinical presentation, remains the most widely used method for sepsis diagnosis (Henriquez-Camacho and Losa, 2014 [9]). Although CRP is widely used in this setting and is considered one of the earliest biomarkers for sepsis, it has low sensitivity (ranging between 30 and 97.2%) and specificity (between 75 and 100%) (Morley and Kushner, 1982 [10]). PCT, the peptide precursor of calcitonin released from the para-follicular cells of the thyroid gland, has also been viewed as a candidate marker for sepsis. Plasma PCT increases by more than 400 times the baseline level in patients with bacterial sepsis (Taylor et al., 2017 [11]). These are the most common well-established biochemical markers used in sepsis diagnosis. An array of cytokines, chemokines, and soluble plasma proteins are less well-established as clinical tools in sepsis management. IL-6 and IL-8 have been shown to play significant roles in sepsis, septic shock, and multiple organ dysfunction (Hou et al., 2015; Matsumoto et al., 2018 [12,13]). Furthermore, sTREM-1, the soluble form of TREM-1, expressed on neutrophils and monocytes and released from activated phagocytes into body fluids, is a promising sepsis biomarker (Jiyong et al., 2009 [14]). Soluble urokinase-type plasminogen activator receptor (suPAR), which mediates adhesion, migration, chemotaxis, and proteolysis of immunological cells (Eugen-Olsen, 2011 [15]), has been implicated as a biomarker for sepsis diagnosis with a sensitivity of 75% and specificity of 72%. (Donadello et al., 2011 [16]). Presepsin is a 13-kDa-cleavage product of the CD14 (cluster of differentiation 14) receptor expressed on macrophages, monocytes, dendritic cells, and neutrophils, and has also been shown to be a promising sepsis biomarker (Chenevier-Gobeaux et al., 2015 [17]). Only PCT, presepsin, suPAR, and sTREM-1, reportedly, have the best diagnostic and prognostic accuracies for sepsis (Larsen and Petersen, 2017) [6]. This study aimed to compare IL-6, IL-8, TREM-1, uPAR, and presepsin with well-established biomarkers of SSI.

## 2. Materials and Methods

Study type and site: This is a cross-sectional hospital-based study conducted at Salmaniya Medical Complex (SMC), in the Kingdom of Bahrain, between 2019 and 2022.

Study participants: A total of 129 critically ill adult patients were diagnosed with severe septic infections (SSIs) and were admitted into the intensive care unit (ICU). Patients were categorized into three groups: (i) patients with sepsis only (Se = 48), (ii) patients with pneumonia only (Pn = 48), and (iii) patients with both Se and Pn (Se/Pn = 33).

Ethical issues: The participants and/or their guardians were informed of the nature of this study before their consent was obtained. The study protocol was approved by the Research and Ethics Committee of the College of Medicine and Medical Sciences, Arabian Gulf University, Kingdom of Bahrain.

Clinical diagnosis: The diagnosis of sepsis was performed based on the criteria of the American College of Chest Physicians/Society of Critical Care Medicine Consensus Conference Committee (ACCP/SCCM) established in 1991 as SEPSIS-1 (Bone et al., 1992 [18]) and in 2001 as SEPSIS-2 (Levy, et al., 2001 [19]), and revised in 2016 (SEPSIS-3) (Singer et al., 2016 [1]). The former was based on systemic inflammatory response syndrome (SIRS) criteria: body temperature above 38 °C or below 36 °C, heart rate greater than 90 beats per minute, respiratory rate greater than 20 beats per minute, and white blood count (WBC) above 12,000/mm^3^ or below 4000/mm^3^ (Bone et al., 1992 [18]). The laboratory diagnostic tools were c-reactive protein (CRP), procalcitonin (PCT), and plasma lactate. The hemostasis markers, platelet count (PLT), prothrombin time (PT), activated partial thrombin time (APTT), international normalized ratio (INR), D-dimer, and fibrinogen, together with organ-damage markers (ODM), bilirubin (liver), and creatinine (kidney), were used as supporting tools. However, the final diagnosis was the practicing physician’s decision.

Blood sampling: Peripheral whole-blood samples were collected from the participants by venipuncture in ethylenediaminetetraacetic acid (EDTA) tubes as part of the routine sampling of similar patients. Blood samples were drawn from patients with sepsis syndrome on the day of diagnosis before treatment. Plasma was separated by centrifugation for 15 min at 1000× *g* (at 2 °C to 8 °C) within 30 minutes of blood collection, aliquoted, and stored at −80 °C until use.

Laboratory investigations and diagnosis: The complete blood count (CBC, including WBC and PLT) and the hemostasis parameters (PT, APTT, fibrinogen, INR, and D-dime) were analyzed in the SMC, the central laboratory, using an automated hemo-analyzer. The inflammation markers (CRP, ESR, and PCT) were measured with different chemo-analyzers. Importantly, the physicians’ final judgment in diagnosing all the sepsis cases was based on the results of these markers plus the blood culture results and other clinical and laboratory findings, including the disease history.

Enzyme-linked immunosorbent assay (ELISA): The levels of cytokines/chemokines and other inflammatory markers were assayed by solid-phase sandwich ELISA using Invitrogen ELISA kits—EH2IL6 (for IL-6), KHC0081 (for IL-8), EHTREM1 (for TREM-1), and EHPLAUR (for uPAR)—following the protocols provided with the kits, as previously described (Sater et al., 2023 [19]). Similarly, presepsin was estimated using Human Presepsin ELISA kit from MyBioSource, Catalog number MBS766136. A Thermo Multiscan Spectrum Plate Reader coupled with SkanIt RE for MSS 2.4.2 software was used for measuring the plates’ absorbances.

Statistical analysis: Sigma Stat software (part of SigmaPlot 15) was used for analysis. Differences between study groups were analyzed by the T-test/Mann–Whitney Rank Sum Test (MW), and ANOVA/one-way analysis of variance/Kruskal–Wallis one-way analysis of variance on ranks (KW). To isolate the group or groups that differed from the others, a multiple comparison procedure, All Pairwise Multiple Comparison Procedures (Dunn’s method), was used. The Pearson Product Moment Correlation was used in the correlation analysis. The statistical significance was set at *p* < 0.05.

Correlation analysis plan: To compare the tested biomarkers, the plasma concentrations of IL-6, IL-8, TREM1, uPAR, and presepsin were assessed using correlation analysis. The levels of each biomarker were correlated with levels of the diagnostic parameters used, the HR, WBC, CRP, PCT, and lactate, in addition to the supporting tests and the hemostasis parameters, PLT, fibrinogen, PT, APTT, INR, and D-dimer. Since sepsis could progress to single- or multiple-organ failure, the liver function parameter, bilirubin, and the renal function parameter, creatinine, were also considered in the analysis as organ dysfunction markers (ODMs). However, the data regarding the respiratory rate (RR) and body temperature (Temp), although used in the diagnosis, were not used in the analysis as explained below. Unfortunately, PaO2 data were not available for several patients.

## 3. Results

### 3.1. Infection, Inflammation, and Clinical Parameters as Diagnostic Criteria for Sepsis

For diagnosis of sepsis, HR, WBC, CRP, PCT, and lactate were used in addition to the physicians’ clinical judgment. Moreover, the hemostasis parameters, PLT, fibrinogen, PT, APTT, INR, and D-dimer were measured to support the diagnosis and disease progress (Table 1). The RR, temperature, and, blood pressure (BP) were excluded from the correlation analysis in this study because many patients were under ventilators, limiting the use of RR as a parameter. At the same time, both high and low temperatures and BPs were frequently recognized among the patients and, therefore, the two parameters could not be used in the correlation analysis.

### 3.2. The Age and Gender Distribution of the Tested Biomarkers

The median (25–75%) ages of the three study groups, i.e., patients with sepsis (Se), pneumonia (Pn), and Se/Pn, were comparable: 58.5 (37.5–68.0), 64.5 (43.25–77.25), and 64.0, (46.5–73.0), respectively, *p* 0.086 KW (Table 1). Importantly, none of the tested biomarkers, i.e., IL-6, IL-8, TREM-1, and presepsin, correlated with age (*p* = 0.733, 0.771, 0.478, and 0.168, respectively), except for uPAR, *p* 0.051, CC 0.181, which was borderline (Figure 1–[I]). Similarly, there were no significant differences in the concentrations of the tested biomarkers between males and females. The plasma concentrations of the test biomarkers, IL-6, IL-8, TREM-1, uPAR, and presepsin, between females and males were not significantly different, with *p*-values of 0.479, 0.312, 0.503, 0.603, and 0.752, respectively (MW) (Figure 1A–E).

### 3.3. Correlations of Tested Biomarkers with the Major Diagnostic Parameters of Sepsis

As seen in (Figure 2), the diagnostic parameters, i.e., HR, WBC, CRP, PCT, and lactate, were best correlated with presepsin, with *p* 0.824, 0.000, 0.910, 0.000, and 0.000, respectively, followed by IL-6, with *p* 0.003, 0.511, 0.806, 0.183, and 0.000, respectively, and IL-8, with *p* 0.002, 0.743, 0.669, 0.196, and 0.000, respectively. The lowest correlations were with the uPAR biomarker, with *p* 0.485, 0.424, 0.695, 0.046, and 0.117, respectively, while TREM1 showed no correlations with any of the diagnostic criteria, with *p* 0.834, 0.540, 0.342, 0.517, and 0.305, respectively.

### 3.4. Correlations of the Tested Biomarkers with the Hemostasis Parameters as Secondary Diagnostic Markers

The tested biomarkers’ correlations with hemostasis parameters are presented in Table 2. The concentrations of IL-6 and IL-8 were positively correlated with PT (*p* > 0.001 and *p* 0.001, respectively), INR (*p* 0.000 and *p* 0.001, respectively), and D-dimer (*p* 0.000 and *p* 0.003, respectively) and negatively correlated with the PLT (*p* 0.017, and *p* 0.020, respectively), but were not correlated with fibrinogen (*p* 0.576, and *p* 0.799, respectively) or APTT (*p* 0.237 and *p* 0.721, respectively). The plasma TREM1 levels were positively correlated with PLT (*p* 0.020) and negatively correlated with APTT (*p* 0.004), but were not correlated with the levels of the remaining four parameters, while the uPAR plasma levels were only positively correlated with D-dimer levels (*p* 0.015). However, the presepsin levels were positively correlated with PT (*p* 0.002), APTT (*p* 0.013), INR (*p* 0.005), and D-dimer (*p* 0.028) and negatively correlated with the PLT levels (*p* 0.004), but were not correlated with fibrinogen (*p* 0.734).

### 3.5. Correlations of the Tested Biomarkers with the Organ Dysfunction Markers

Two indicators for organ damage, bilirubin for the liver and creatinine for the kidney, were measured. The data for the latter were available for only a limited number of patients (48–56). The bilirubin was significantly positively correlated with IL-6 (*p* 0.000, CC 0.465), IL-8 (*p* 0.002, CC 0.288), and presepsin (*p* 0.000, CC 0.368), while it was not correlated with TREM-1 (*p* 0.208, CC−0.118) or uPAR (*p* 0.126, CC 0.143). On the other hand, the plasma creatinine levels were correlated only with presepsin (*p* 0.000, CC 0.521) and uPAR (*p* 0.024, CC 0.269) levels, but not with IL-6 (*p* 0.290, CC 0.134), IL-8 (*p* 0.462, CC0.0936), or TREM-1 (*p* 0.096, CC −0.198).

### 3.6. Comparisons of Serum Levels of Test Biomarkers in the Clinical Types of SSI, Sepsis (Sp), Pneumonia (Pn), and Mixed Se/Pn

The three subgroups of severe septic infections, sepsis (Se), pneumonia (Pn), and both Se/Pn, were comparable in age and sex distribution, and also had comparable levels of the primary diagnostic parameters, except for WBC, were significantly higher in Se (17.02, 12.49–24.39) compared to both Pn (13.43, 9.04–17.61) and Se/Pn (13.40, 9.03–16.71) patients, *p* 0.015 and 0.018, respectively (Table 1). As seen in Figure 3, the median plasma concentrations of IL-6 and TREM1 in the Se, Pn, and Se/Pn groups were comparable (*p* 0.581 and *p* 0.374, respectively) (KW). The plasma concentration of IL-8 was significantly different between the Se, Pn, and Se/Pn groups; the levels were significantly higher in Se compared to the other 2 subgroups, *p* < 0.001, in both comparisons (KW). Similarly, the plasma level of presepsin was significantly higher in the Se group compared with the Pn group, *p* 0.00, and the Se/Pn group, *p* < 0.001. On the contrary, the plasma levels of uPAR were significantly lower in the Se group compared to the Pn, *p* 0.00, and the Se/Pn, *p* 0.00, groups.

### 3.7. Comparisons of Serum Levels of Test Biomarkers Between SSI Patients Based on Severity Markers

Since we did not have any clinical or laboratory evidence for the severity grading in this setting, we examined the use of artificial ventilation (AV) as an indicator of severity. We found that 64 patients were put under ventilators (AV patients) during the study while 63 patients were not (non-AV patients), and 2 patients were not known to be artificially ventilated or not. However, when we compared the levels of each of the five tested biomarkers in the two groups (AV vs. non-AV patients), the only biomarker that was found to have significantly different levels between the groups was TREM-1 (−357.15,−494.43–−67.45 vs. −256.87, −425.82–−401.86, respectively, *p* 0.041), which was higher in the patients not under the ventilator. In contrast the levels of the presepsin (3.72, 1.55-13.00 vs. 4.74, 1.156-10.70, respectively, *p* 0.703), uPAR (2015.11, 1434.97-2796.14, vs. 2007.10, 1056.45-2922.40, respectively, *p* 0.615), IL-6 (31.14, 4.96-99.89 vs. 14.72, 0.45-63.66, *p* 0.081), and IL-8 (−79.66, −103.80-−27.74 vs. −91.89, −104.99-−60.69, respectively, *p* 0.294), were comparable between the two groups. Unexpectedly, all the diagnostic parameters—HR, WBC, CRP, PCT, and lactate—were also comparable between the two groups.

## 4. Discussion

The present study aimed to compare the clinical utility of sepsis biomarkers, identify the ones that could potentially consolidate diagnosis, differentiate the types and severity of sepsis, and possibly detect the associated organ damage using correlations and comparative analysis versus clinical and diagnostic markers. The diagnostic properties, including sensitivity, specificity, and hazard ratios, have been thoroughly investigated previously (Pierrakos C, Vincent JL, 2010 [20]); therefore, they were not included in this study. The five tested biomarkers, IL6, IL8, TREM-1, suPAR, and presepsin, have previously been shown to be remarkable biomarkers of sepsis with pneumonia (Larsen and Petersen 2017 [16]). The correlations of the tested biomarkers with a. the current diagnostic parameters, b. hemostasis markers, c. SSI-associated organ damage markers, and d. the severity grading parameter of SSI, in addition to e. the distinction between the SSI subgroups, are expected to pave the way for selection of a smaller number of markers of wider use in clinical practice. In the discussion, the term “biomarkers” is reserved for the tested ones to distinguish them from the diagnostic, hemostasis, or other markers.

In this setting, neither sex nor age was found to influence the tested biomarker levels, and they were comparable between the subgroups of SSI (Table 1), although both factors have previously been found to be compelling risk factors in sepsis (Ko et al., 2023 [21]). However, this study was not designed to show the role of age in SSI development. A major finding of the current study was that presepsin, compared to IL-6, IL-8, TREM-1, and uPAR, and the conventional diagnostic parameters, i.e., HR, WBC, CRP, PCT, and lactate, showed the highest quantitative correlation scores with all markers of SSI, including diagnostic, hemostasis, and ODM. Also, presepsin was the best marker to discriminate between the SSI types. This might qualify presepsin for an upgrade into a diagnostic criterion, supporting the previously reported meta-analysis of 129 studies (Yoon et al., 2019 [22]) as well as other studies (Ulla et al., 2013 [23]). In addition, in the current study, the CRP was the only diagnostic marker that was found to be abnormally raised in all patients (diagnostic rate 100%), supporting other studies (Koozi et al., 2019 [24]), unlike the other diagnostic markers, whose diagnostic rates varied between 79% for HR and 33.8% for lactate (Table A1 in Appendix A). In contrast, the CRP showed no correlation with the hemostasis parameters or ODM; therefore, CRP cannot replace the other markers used in prognostication or differentiating SSI types (Koozi et al., 2019 [24]). Moreover, the CRP levels had no correlations with any of the tested biomarkers, unlike the HR and lactate, which correlated with the levels of 3 out of 5 biomarkers (Figure 2). This may be interpreted as proving that CRP is most useful for the diagnosis of sepsis and the least for grading sepsis severity among the diagnostic criteria utilized, as reported previously (Henriquez-Camacho and Losa, 2014 [9]), unlike lactate (Kang and Park, 2016 [25]). Therefore, it is imperative to use several diagnostic markers for SSI management, since there is no inclusive biomarker for sepsis management.

In sepsis, infections and inflammations are mistakenly used synonymously, although they are not mutually exclusive; however, their markers, e.g., cytokines, chemokines, presepsin, and TREM-1, are shared. Infections are usually associated with inflammation, while the opposite is not true, as several pathologies other than infection could elicit inflammation, especially in the older population, which is the most vulnerable to sepsis (Ibarz et al., 2024 [26]). This adds a level of complexity; therefore, biomarkers more associated with infection than inflammation had a discriminative advantage over the inflammatory markers, e.g., IL-6, IL-8, CRP, PCT, lactate, TREM-1, and uPAR.

We ran correlations between the five tested biomarkers with a total of 13 analyzed diagnostic and prognostic parameters and found that presepsin showed the highest rate of significant correlations with the diagnostic parameters (3 of 5; WBC, PCT, and lactate), followed by IL-6 and IL-8 (2/5; HR and lactate) and uPAR (1/4, PCT), while TREM-1, unexpectedly, was not correlated with any diagnostic parameters. The observation is in line with previous studies that have shown that the levels of presepsin increase significantly in local infection, SIRS, and sepsis (Shozushima et al., 2011; Endo et al., 2012 [27,28]). Also, the levels of presepsin were the most correlated with levels of the supporting diagnostic hemostasis parameters (5/6, all markers except fibrinogen), followed by IL-6 and IL-8, with scores of 4/6 (all except fibrinogen and APTT), followed by TREM-1, which scored 2/6 (PLT and APTT), while uPAR (1/6) was only correlated with D-dimer (Table 2). It is well known that there is cross-talk between inflammation and coagulation; thus, coagulopathy is common in sepsis, potentially worsening the prognosis (Tsantes et al., 2023 [29]). With regard to the organ damage markers, presepsin was the only biomarker that correlated with bilirubin levels as a marker for liver injury, while presepsin and uPAR levels were the only biomarker levels that correlated with creatinine as a signal for kidney damage. This further corroborates the leading role of presepsin as a biomarker for SSI progression, prognosis, and prediction of SSI-associated organ damage. It was previously mentioned that presepsin levels could differentiate sepsis from non-infectious organ failure and help clinicians to identify sepsis patients with a poor prognosis (Lee et al., 2022 [30]). It is worth noting that all the correlations of the presepsin were positive ones, i.e., as the level of the tested parameter increased, the level of the presepsin increased, except with PLT, since the latter decreased when the presepsin levels increased, i.e., a negative correlation, as stated before (Tsantes et al., 2023 [29]). Upon ranking the five tested biomarkers out of a total score of 13, presepsin scored 10/13, followed by IL-6 and IL-8 (7/13), then uPAR (3/13), and the least correlated levels were found for TREM-1 (2/13).

Moreover, presepsin, together with IL-8 and uPAR, showed better discrimination between the subgroups of SSI, i.e., Se, Pn, and Se/Pn (Figure 3). This is important in grading the disease severity, prognosis, and identification of the exact cause of the SSI, as reported previously (Galliera et al., 2019 [31]; Stankovic, 2022 [32]). Interestingly, all the diagnostic parameters used showed no discriminative differences in their levels between the three clinical entities, except the WBC (Table 1), which was significantly higher in Se compared to the Pn and Se/Pn, similar to the presepsin and IL-8, but unlike the uPAR, which was significantly higher in the combined Se/Pn infection group than in Pn and Se. However, whether Se/Pn is more severe than the other two, or vice versa, remains unknown from a clinical point of view, as there is no evidence that the grading of severity was different between the three subgroups. This role of WBC further supports the idea of having multiple diagnostic parameters in cases of sepsis (Singer et al., 2016 [1]; WHO, 2023 [2]). On the other hand, three of the six supportive diagnostic parameters of hemostasis, the PT, APTT, and INR, were shown to discriminate between the clinical groups of SSI, with similar patterns to the presepsin, IL-8, and WBC, which were higher in the Se group compared to the other two groups. As an inference, while the diagnostic criteria used herein failed to distinguish between the clinical groups, which is decisive for treatment, presepsin and the other two markers were able to be discriminative. However, this needs to be considered with caution, as an explanation is required as to the reason for the higher levels present in Se but not in the combined Se/Pn group. In other studies, the correlation between presepsin initial values and in-hospital mortality has suggested that this biomarker could be used for early, reliable risk stratification and to identify high-risk patients (Ulla et al., 2013 [23]).

Finally, due to the lack of well-defined indicators of severity other than the clinical ones, we assumed that use of AV indicated a more severe status. Of the five tested biomarkers, only the levels of TREM-1 (*p* 0.041) were found to be significantly different between the patients on AV and the others. Unexpectedly, the TREM-1 levels were higher in the patients who were not under AV. A possible explanation was that the AV improved the patients’ statuses compared to the other patients. The inference from this result is that there is an urgent need for an objective and measurable criterion for grading the severity of SSI, as all the diagnostic parameters also failed to distinguish between the two groups. A simple system for grading the severity of sepsis was developed in 1983 by scoring the attributes of sepsis under four headings: local effects of infection, pyrexia, secondary effects of sepsis, and laboratory data (Elebute and Stoner, 1983 [33]); however, the version updated in 2017 (Khwannimit et al., 2017 [34]) included mechanical ventilation as a parameter.

Although the overall rating of the tested biomarkers in terms of correlations with clinical and diagnostic parameters and, thus, stratification of the SSI and discrimination between its types approved the preeminence of presepsin, the other tested biomarkers were found to fill the gaps where presepsin was lagging or to support the presepsin results. The IL-6 and IL-8 ranked second in correlations with the diagnostic, hemostasis, and ODM markers in SSI among the tested biomarkers that were scored (7/13). However, IL-8 was found to further discriminate between the clinical groups, while both interleukins had no role in severity grading, as judged by the use of AV. The IL-6 and IL-8 biomarkers have previously been reported to have the next best predictive ability for sepsis after PCT (Harbarth et al., 2001 [35]), in line with a systematic investigation approving the usefulness of IL6 and IL8 as sepsis markers (Hou et al., 2015 [12]; Matsumoto et al., 2018 [13]). However, there are limitations to the clinical utility of both biomarkers, as they are not specific to sepsis and can be raised in several other acute and chronic non-infectious inflammatory disorders, e.g., diabetes mellitus type 2 (Sater et al., 2023 [19]); Rohm et al., 2022 [36]), which are frequent in patients with sepsis.

The TREM-1 and uPAR were rated as the third and lowest (score 2/13) in our correlation rankings of SSI diagnosis; however, uPAR was found to be more advantageous in discrimination between the SSI clinical groups, while TREM-1 had the credit of being a marker of severity (AV), as it was the only biomarker that distinguished the patients under ventilators. Other studies reported the practical validation of TREM-1 and uPAR and their promising roles in sepsis diagnosis (Jiyong et al., 2009 [14]; Donadello et al., 2011 [16]).

In this study, none of the diagnostic markers used were discriminative between the SSI types except WBC, as mentioned above. The CRP was the only diagnostic parameter that was not correlated with the levels of any of the tested biomarkers, in contrast to lactate, which was the most correlated one (Figure 2). This observation highlights the importance of lactate as a diagnostic marker of sepsis, as mentioned elsewhere (Kang and Park et al., 2016 [25]; Garcia-Alvarez et al., 2014 [37]). In contrast, the CRP was the only diagnostic marker that was raised above the normal reference range in all study subjects. This might indicate that the CRP was the gold-standard test on which the clinical diagnosis of sepsis was based in this setting, but it was less important in grading the cases of sepsis, as reported before (Luzzani et al., 2003 [38]).

Finally, as a promising biomarker, it is worth highlighting the potential of presepsin as a diagnostic parameter and knowing about its biological role to explain the importance of presepsin in sepsis diagnosis. Plasma presepsin levels are very low in healthy subjects and have been shown to increase fast and sharply in response to bacterial infections according to the severity of the disease (Chenevier-Gobeaux et al., 2015 [17]; Piccioni et al., 2021 [39]). The specificity and uniqueness of presepsin as a sepsis biomarker are derived from its natural biological function as an innate immune response in bacterial infection. As mentioned, presepsin is produced from sCD14 by cleavage of the N-terminal (Chenevier-Gobeaux et al., 2015 [17]). The CD14 acts as a coreceptor for various bacterial ligands. For example, the lipopolysaccharide (LPS) as a gram-negative bacterial ligand can associate with LPS-binding protein (LBP), CD14, and toll-like receptor-4 to form a complex on effector cells like tissue monocytes/macrophages. This interaction mediates intracellular signaling and activates the production of cytokines, which is the initial host inflammatory response against the pathogen (Chenevier-Gobeaux et al., 2015 [17]), acting as an immune ignition response.

Currently, presepsin can be measured precisely by chemiluminescence enzyme immunoassays using automated analyzers (Yaegashi 2005 [40]; Shirakawa et al., 2011 [41]) at a cost-effective price. Moreover, it has been shown to elevate within 2 h and peak at around 3 h following the onset of infection (Piccioni et al., 2021 [39]). Therefore, this study strongly supports a previous recommendation that presepsin should be genuinely investigated for use in clinical practice for sepsis diagnosis (Mussap et al., 2011 [42]; Mussap et al., 2012 [43]).

The limitations of this study include the ethical issues raised in dealing with critically ill patients in terms of recruitment of patients and the sampling of blood from the recruited ones. Also, the nature of sepsis as a cause of multi-organ failure, adds more confounding factors. Moreover, in the cross-sectional observational study, the deployment of the biomarkers for prognosis assessment and pathophysiology study was unlikely to be sufficient; therefore, longitudinal follow-up studies are required. In conclusion, in this study, compared with the other tested SSI biomarkers, i.e., IL6, IL8, sTREM1, and suPAR, presepsin showed the highest score in supporting the diagnosis of SSI, discriminating between the clinical types of SSI and predicting SSI-associated organ dysfunction as well as the hemostasis status in these patients. The correlations of presepsin levels with the levels of most of the aforementioned sepsis markers, unlike all other tested biomarkers and markers (diagnostic, supportive, and ODM markers), qualify presepsin to be a leading diagnostic tool, with the advantage of being more specific to infection than to other non-infectious inflammatory illnesses, as well as having a quick turnaround time and cost-effectiveness. However, there is still room to consider other biomarkers to complement the diagnostic tools for specific goals as part of the set of markers.

## Figures and Tables

**Figure 1 diagnostics-15-00217-f001:**
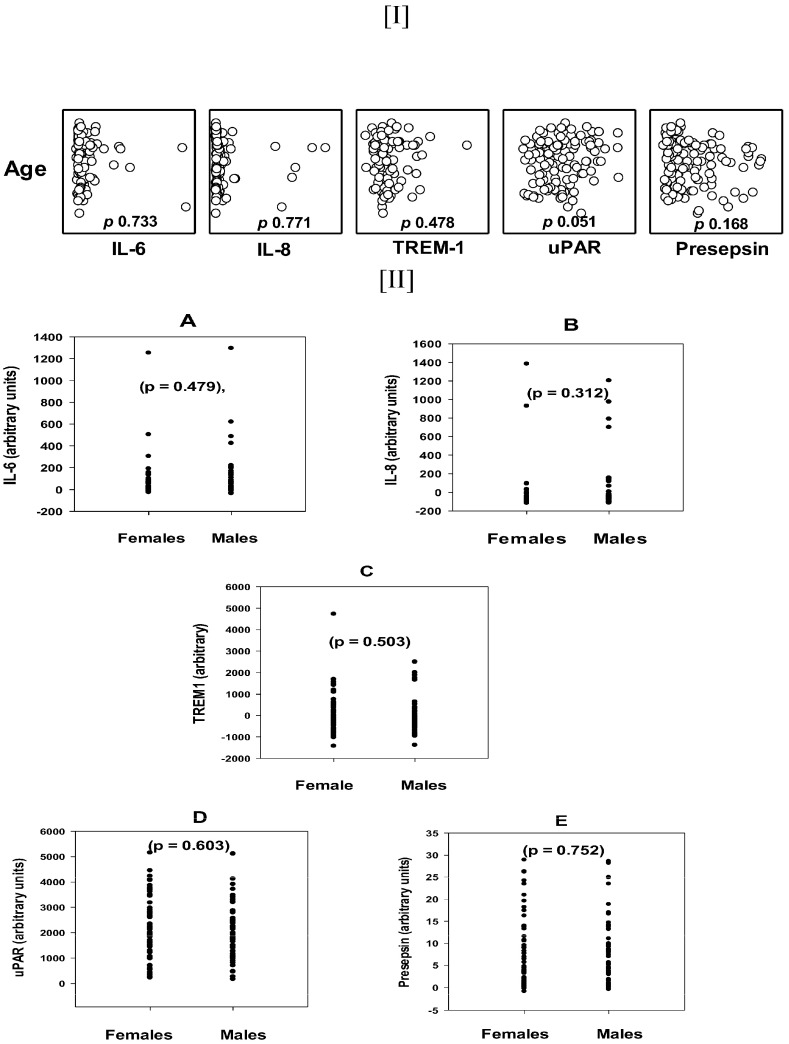
The effect of age and sex on the plasma levels of the tested biomarkers: IL-6, IL-8, TREM-1, uPAR, and presepsin. Upper figure [I]: scatter plot shows the correlations of the plasma levels of the biomarkers versus age. *p*-values are shown in the figure. The only borderline correlation was of the uPAR levels (*p* 0.051, CC (correlation coefficient) 0.181). Note: the open circles stand for each study subject separately. Lower figure [II]: plasma levels of biomarkers, comparisons between males and females. *p* values are placed in the figure. The plasma levels of the tested biomarkers; IL-6 (**A**), IL-8 (**B**), TREM-1 (**C**), uPAR (**D**), and presepsin (**E**), for each individual (black dots) are shown in the figure separately. However, for all study subjects, taken together the plasma median (75th and 25th percentile) concentrations (arbitrary units) of the above test biomarkers, as compared between females and males were 23.534, 3.865–59.393 vs. 24.825, 2.534–118.155 (*p* 0.479); −94.659, −104.036–−64.762 vs. −79.663, −104.929–−42.497 (*p* 0.312); −271.631, −455.717–209.744 vs. −364.803, −463.566–−23.093 (*p* 0.503); 2135.601, 1218.612–2953.504 vs. 1838.068, 1213.172–2821.403 (*p* 0.603); and 3.865, 1.156–10.704 vs. 4.459, 1.595–11.567 (*p* 0.752), respectively (MW). No significant differences were found in the levels of the biomarkers between both sexes.

**Figure 2 diagnostics-15-00217-f002:**
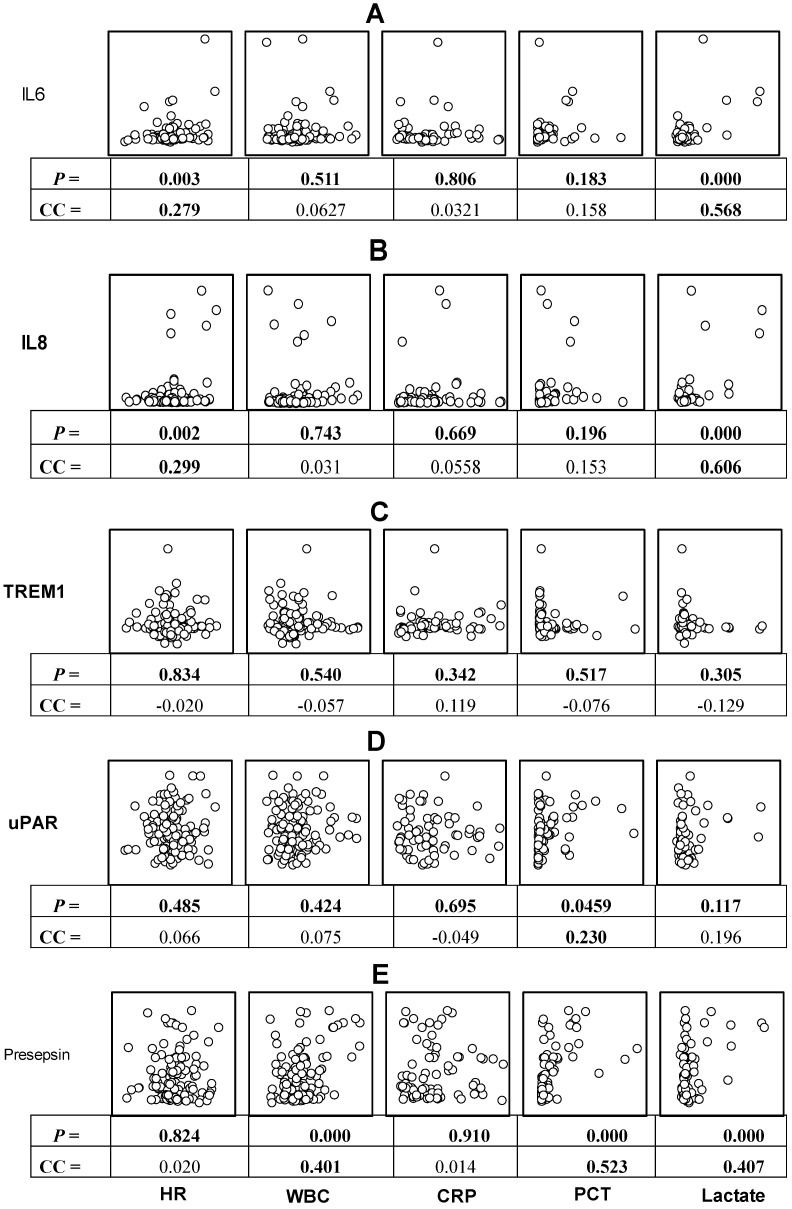
Scatter plot showing the correlations of plasma levels of each of the tested biomarkers, IL-6 (**A**), IL-8 (**B**), TREM-1 (**C**), uPAR (**D**), and presepsin (**E**), versus the five diagnostic parameters, heart rate (HR), white blood count (WBC), c-reactive protein (CRP), procalcitonin (PCT), and lactate. The Il-6 and IL-8 levels were significantly positively correlated with the HR and lactate levels, the TREM-1 levels were not correlated with any diagnostic parameter, and uPAR levels were positively correlated with PCT, while presepsin levels were significantly positively correlated with WBC, PCT, and lactate. The *p*-values and CC (correlation coefficient) are shown in the figure. Notably, the CRP levels were not correlated with any of the tested biomarkers.

**Figure 3 diagnostics-15-00217-f003:**
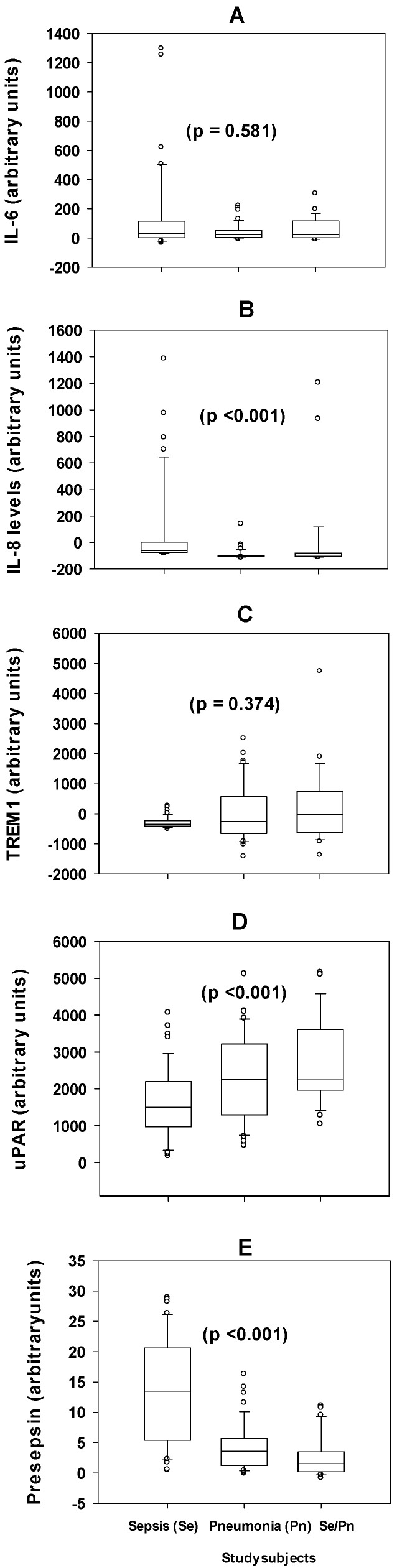
The median plasma concentrations (arbitrary units) of (**A**) IL-6 in sepsis (Se) (32.26, 1.66–114.79), pneumonia (Pn) (22.56, 3.81–53.97), and Se/Pn (24.49, 2.31–116.24) groups were comparable, *p* 0.581 (KW). (**B**) The plasma concentrations of IL-8 in the Se (−60.397, −73.872–3.775), Pn (−102.684, −108.145–−94.659), and Se/Pn (−101.943, −106.794–−78.764) groups were different, *p* < 0.001 (KW). (**C**) The plasma concentrations of TREM1 in the Se (−349.10, −419.13–−236.89), Pn (−257.80, −654.24–563.93), and Se/Pn groups (−33.75, −618.46-739.41) were comparable, *p* 0.374 (KW). (**D**) The plasma levels of uPAR in the Se (1507.221, 976.191–2200.685), Pn (2259.057, 1293.304–3220.516) and Se/Pn (2242.496, 1964.184–3619.263) groups were different, *p* 0.000. (**E**) The plasma levels of presepsin in the Se (13.464, 5.393–20.599), Pn (3.548, 1.213–5.674), and Se/Pn (1.545, 0.214–3.475) groups were different, *p* < 0.001. The horizontal line within each bar is the median value; the bottom and top lines of the bar are 25% and 75%, respectively; caps of the lower and upper vertical lines are the 5% and 95% percentiles; and the open circles are outliers.

**Table 1 diagnostics-15-00217-t001:** Description and comparison of the clinical and laboratory parameters between study subjects, with three types of severe septic infection (SSI).

Variables	Sepsis (Se) Patients	Pneumonia (Pn) Patients	Se/Pn Double Infection Patients	*p* Value
Number	48	48	33	
Sex (M/F—ratio)	23/25	23/25	16/17	
Age (years)	58.5, 37.5–68.0	64.5, 43.25–77.25	64.0, 46.5–73.0	0.086
Vital functions, Temp and Blood count				
HR (bpm)	102.0, 93.25–117.5	98.0, 90.5–114.75	104.0, 94.0–113.0	0.608
RR	24.5, 21.25–30.0	26.0, 22.75–29.0	32.0, 28.0–38.5	0.049
Temp (°C)	36.95, 36.05–38.1	36.95, 36.5–37.88	38.0, 36.5–38.45	0.115
WBC (×10^9^/L)	17.02, 12.49–24.39	13.43, 9.04–17.61	13.4, 9.03–16.71	0.019
Inflammatory markers				
CRP (mg/L)	100.0, 42.3–144.0	82.4, 24.2–216.5	96.1, 63.4–137.5	0.992
PCT (mg/L)	3.53, 0.5–15.6	1.21, 0.412–2.58	1.83, 0.302–7.4	0.150
Lactate (mmol/L)	1.85, 1.375–3.525	NK	1.7, 1.35–2.2	0.35
Hemostasis parameters				
Platelets (×10^9^/L)	192.0, 93.5–275.75	255.5, 169.0–328.0	199.0, 127.5–367.5	0.052
PT (sec)	15.25, 13.53–18.28	12.9, 12.0–15.08	13.9, 12.85–15.45	<0.001
APTT (sec)	27.5, 23.4–39.0	23.2, 21.0–30.8	25.9, 21.05–29.50	0.032
INR	1.325,1.155–1.578	1.100, 1.012–1.282	1.180, 1.105–1.305	<0.001
Fibrinogen (g/L)	403.88 ± 151.49	455.68 ± 194.99	458.08 ± 200.25	0.355 ANOVA
D-dimer (mg/L)	5.67, 1.85–10.08	3.42, 1.93–11.44		0.631 MW
Organ dysfunction parameters Severity indicators				
Bilirubin (µmol/L)	13.5, 7.25–44.0	12.0, 9.0–22.0	15.0, 8.5–28.0	0.508
Creatinine (µmol/L)	129.0, 54.8–267.8	78.0, 38.0–160.0	NK	0.076 MW
Artificial ventilation	54.5% (24/44)	43.5% (20/46)	60.6% (20/33)	0.296 Chi-square

Note: All statistical tests were carried out by KW unless otherwise mentioned in the last column. BPM = beats per minute; sec = second; NK = not known, HR = heart rate, RR = respiratory rate, BP = blood pressure, WBC = white blood count, CRP = C-reactive protein, PCT = procalcitonin, PLT= platelets count, PT = prothrombin time, APTT = activated partial thromboplastin time, INR = international normalization ratio.

**Table 2 diagnostics-15-00217-t002:** Correlations of the plasma levels of test biomarkers with hemostasis parameters in all study subjects taken together.

Test Biomarker		Platelets Count	Fibrinogen	PT	APTT	INR	D-Dimer	Score
Presepsin	*p*	0.004	0.734	0.002	0.013	0.005	0.0284	5
	CC	−0.250	0.0357	0.265	0.221	0.248	0.291	
IL-6	*p*	0.017	0.576	0.000	0.237	0.000	0.000	4
	CC	−0.225	0.0639	0.581	0.114	0.573	0.550	
IL-8	*p*	0.020	0.799	0.001	0.721	0.001	0.003	4
	CC	−0.219	0.0291	0.321	0.0346	0.320	0.418	
TREM-1	*p*	0.020	0.691	0.118	0.004	0.147	0.733	2
	CC	0.214	0.0438	−0.145	−0.270	−0.135	−0.047	
uPAR	*p*	0.229	0.984	0.591	0.171	0.607	0.0146	2
	CC	−0.112	−0.0022	0.0502	0.129	0.0481	0.328	

Note: The light grey shaded cells highlight statistically significant (*p* < 0.05) positive correlations, while the darker ones highlight the significantly negative correlations (CC = correlation coefficient). WBC = white blood count, PLT = platelets count, PT = prothrombin time, APTT = activated partial thromboplastin time, INR = international normalization ratio.

## Data Availability

Raw data were generated at the Arabian Gulf University. Derived data supporting the findings of this study are available from the corresponding author (M.S.S.) upon request.

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
