# Peer review of "Potentials of Presepsin as a Novel Sepsis Biomarker in Critically Ill Adults: Correlation Analysis with the Current Diagnostic Markers"

_diagnostics, 2025, doi:10.3390/diagnostics15020217_

Round 1
Reviewer 1 Report
Comments and Suggestions for Authors
I have finished the paper titled “Potentials of Presepsin as a Novel Sepsis Biomarker in Critically Ill Adults: Correlation Analysis with the Current Diagnostic Markers” . From front to back, the author's writing is quite good but also somewhat inadequate in my opinion. For example, SIRS diagnostic criteria on page 118 you only include respiratory rate and actually should have blood gas analysis the level of Pa CO2. Secondly, there is no analysis of the biological role of Presepsin in the discussion. It is best to have a short analysis and explanation the biological function of Presepsin. Other aspects are suitable.
Comments on the Quality of English LanguageThe quality of English l is slightly well
Author Response
Comments and Suggestions for Authors
Comments on the Quality of English Language: The quality of English l is slightly well
Response: We did intensive editorial work on the manuscript including English revision and corrections
Comment: For example, SIRS diagnostic criteria on page 118 you only include respiratory rate and actually should have blood gas analysis the level of Pa CO2.
Response: We totally agree with the importance of PaCO2 at least from a clinical point of view but unfortunately, the Pa CO2 data, was not available for several patients and therefore dropped from the analysis. However, we added a paragraph in the amended manuscript explaining that aspect.
Comment: Secondly, there is no analysis of the biological role of Presepsin in the discussion. It is best to have a short analysis and explanation of the biological function of Presepsin. Other aspects are suitable.
Response: A paragraph in the discussion was added to explain the biological role of presepsin in infection. We mentioned that the specificity and uniqueness of presepsin as a sepsis biomarker are derived from its natural biological function as an innate immune response in bacterial infection. The presepsin is produced from soluble CD14 (sCD14) by the cleavage of the N-terminal. The CD14 acts as a coreceptor for various bacterial ligands. For example, the lipopolysaccharide (LPS) as a gram-negative bacteria ligand can associate with LPS-binding protein (LBP), CD14, and toll-like receptor-4 to form a complex on effector cells like tissue monocytes /macrophages. That interaction mediates intracellular signaling and activates the production of cytokines, which is the initial host inflammatory response against the pathogen. Thus, presepsin stands as a gate for the innate immune system to the invading pathogens.
Reviewer 2 Report
Comments and Suggestions for Authors
I reviewed of Manuscript: "Potentials of Presepsin as a Novel Sepsis Biomarker in Critically Ill Adults".
This study is important because there is still the need to find the best biomarkers for sepsis and its management in clinical practice. Analysis of presepsin and other biomarkers in relation to conventional diagnostic markers is thoroughly investigated in this study. Presepsin has a high diagnostic and prognostic value and the values are related to clinical and laboratory variables. Biomarkers in the study are related to organ failure and coagulation and, therefore, reflect the pathophysiology of sepsis. The findings also suggest that presepsin can be useful as a simple and inexpensive method of diagnosis in a clinical setting.
I therefore recommend that the paper can be published in the actual form.
Author Response
Comment: I therefore recommend that the paper can be published in the actual form.
Response: We thank you for your time and efforts in reviewing and recommending our paper in actual form.